# Metabolic Profiling Reveals That the Olfactory Cues in the Duck Uropygial Gland Potentially Act as Sex Pheromones

**DOI:** 10.3390/ani12040413

**Published:** 2022-02-09

**Authors:** Hehe Liu, Zhao Yang, Yifa He, Qinglan Yang, Qian Tang, Zhenghui Yang, Jingjing Qi, Qian Hu, Lili Bai, Liang Li

**Affiliations:** Farm Animal Genetic Resources Exploration and Innovation Key Laboratory of Sichuan Province, College of Animal Science and Technology, Sichuan Agricultural University, Chengdu 613000, China; liuee1985@sicau.edu.cn (H.L.); 2021302173@stu.sicau.edu.cn (Z.Y.); 201802193@stu.sicau.edu.cn (Y.H.); yangqinglan@stu.sicau.edu.cn (Q.Y.); 2020302132@stu.sicau.edu.cn (Q.T.); yangzhenghui@stu.sicau.edu.cn (Z.Y.); 2020302126@stu.sicau.edu.cn (J.Q.); 2020302157@stu.sicau.edu.cn (Q.H.); bailili1985@sicau.edu.cn (L.B.)

**Keywords:** duck, uropygial gland, gender, volatile substance, metabolome

## Abstract

**Simple Summary:**

For birds, the uropygial gland is a special organ. We believe that its secretion can be used as a pheromone between the sexes to play a role in mate selection and mating. Therefore, we studied the chemical composition of duck uropygial gland secretions and the differences between males and females. After a series of screenings, 24 different volatile metabolites were obtained in our experiment. On this basis, five extremely significant volatile metabolites were screened out—significantly more males than females. The results show that these volatile substances are potential sex pheromone substances, which may be the critical olfactory clues for birds to choose mates. Our results lay the foundation for further research on whether uropygial gland secretion affects duck reproduction and production.

**Abstract:**

The exchange of information between animals is crucial for maintaining social relations, individual survival, and reproduction, etc. The uropygial gland is a particular secretion gland found in birds. We speculated that uropygial gland secretions might act as a chemical signal responsible for sexual communication. We employed non-targeted metabolomic technology through liquid chromatography and mass spectrometry (LC-MS) to identifying duck uropygial gland secretions. We observed 11,311 and 14,321 chemical substances in the uropygial gland secretion for positive and negative ion modes, respectively. Based on their relative contents, principal component analysis (PCA) showed that gender significantly affects the metabolite composition of the duck uropygial gland. A total of 3831 and 4510 differential metabolites were further identified between the two sexes at the positive and negative ion modes, respectively. Of them, 139 differential metabolites were finally annotated. Among the 80 differential metabolites that reached an extremely significant difference (*p* < 0.01), we identified 24 volatile substances. Moreover, we further demonstrated that five kinds of volatile substances are highly repeatable in all testing ducks, including picolinic acid, 3-Hydroxypicolinic acid, indoleacetaldehyde, 3-hydroxymethylglutaric acid, and 3-methyl-2-oxovaleric acid. All these substances are significantly higher in males than in females, and their functions are involved in the reproduction processes of birds. Our data implied that these volatile substances act as sex pheromones and may be crucial olfactory clues for mate selection between birds. Our findings laid the foundation for future research on whether uropygial gland secretion can affect ducks’ reproduction and production.

## 1. Introduction

Animals communicate through information exchange, which is crucial for maintaining social relations, individual survival, and reproduction, etc. These kinds of information communication can be classified into olfactory (chemical) communication [1], auditory communication [2], visual communication [3], tactile communication, and electrical communication, etc. For visual communication, the visual acuity of birds is excellent and extends to ultraviolet. Birds, such as peacocks, often use bright feather colors to perform elaborate sexual displays [4]. For olfactory communication, studies have shown that in quails, the deprivation of olfactory inputs decreases the neuronal activation induced by sexual interactions with a female. Different bird groups, such as petrels, auklets, and ducks, have been shown to produce special odors, which may play an important role as pheromones in within-species social interactions [4].

The chemical compounds emitted by animals, and which are used in mate attraction and mate choice, are called pheromones [5]. Sex pheromones are substances released by animals of both sexes to identify each other. Through this substance, females and males can approach each other, thereby, leading to mating. Generally, females secrete sporadic sex pheromones to induce sexual excitement in active males, but there are also species in which sex pheromones are secreted by males. I nsects are also sensitive to specific sex pheromones. Based on this, the sex attractant has been developed as a high-tech bionic product that simulates the insect sex pheromone in nature and is released to the field through a releaser to trap and kill heterosexual pests [6]. Some insectivorous birds can exploit the pheromones emitted by female moths to attract males as a method of prey detection [7]. In the terrestrial salamander (*Plethodon shermani*) [8], a male delivers proteinaceous pheromones to the female as part of their ritualistic courtship behavior. These pheromones increase the female’s receptivity to mating, as shown by the reduction in courtship duration. Human pheromones also have sex distinctions; the male pheromone is androstenedione [9], and the female pheromone is estradiol [10].

Many animals have a particular organ that secretes special odors, such as the glandular sac of the male musk deer, which can secrete musk to attract females [11]. Civets have a scent gland in the perineum that secretes civet scent, which plays a role in marking territory and attracting the opposite sex [12,13]. For birds, the uropygial gland is a special skin derivative and hole plasma secretory gland. The uropygial gland is usually arranged in pairs, parallel to the back of the tail feather base, and located under the skin. As the largest exocrine gland in birds, the uropygial gland usually secretes lipids and aliphatic monoesters, which are likely to be essential sources of chemical signals for birds. Using gas chromatography–mass spectrometry (GC-MS), Burger et al. identified the uropygial gland secretion of greenwood hoopoe (*Upupa epops*). They showed the volatile substances consisted of short-chain fatty acids, aldehydes, aliphatic and heterocyclic aromatic amines, ketones, and dimethyl sulfides [14]. Researchers have detected gender differences in the chemical composition of the uropygial gland waxes in domestic ducks before the nesting period [15]. Studies have also found that the volatile substances secreted by the uropygial glands are related to the behavior [16] and reproductive activities of avian species [17]. Additionally, it has been proved that the presence of a fatty acid mixture (called soothing pheromones) in the uropygial gland secretions of hens has a soothing effect on chicks, reducing their stress, anxiety, and aggressive behavior, and promoting the growth of chicks [18]. Furthermore, studies have explained the biological functions of uropygial glands in mate selection [19], production performance [20], and reproductive performance [21,22], etc. Research shows that a passerine species can discriminate the sex of conspecifics by relying on chemical cues, which suggests that uropygial gland secretion may potentially function as a chemical signal used in mate choice and/or intrasexual competition in this species [23]. Hirao et al. found that roosters are more likely to mate with hens with uropygial glands when compared to hens with uropygial glands removed. This means that the uropygial glands of hens can emit some odor information, and male chickens can distinguish by smell [19].

Although growing evidence supported that those chemical compounds emitted by the uropygial gland of birds may play a role in individual recognition, the possible role of chemical cues in the sexual selection of birds has only been preliminarily studied. Waterfowl have well-developed uropygial glands [24]. Research has shown that male domestic ducks with the olfactory nerve removed exhibited significantly inhibited sexual behavior, implying that chemical signals may play a role in duck courtship behaviors [25]. We hypothesized that uropygial gland secretions might act as a chemical signal responsible for sexual communication in ducks. Considering that high-performance liquid chromatography (HPLC) is a chromatographic technique with strong versatility and analytical capabilities, it is suitable for any compound with solubility in liquids and is widely used to quantify small molecules and ions, as well as the separation and purification of large molecules [26]. Therefore, it is necessary to give a comprehensive view of the chemical compounds secreted by the uropygial glands of ducks based on non-targeted metabolomics of LC-MS, and to show the differences of ducks between the sexes to further explore whether there were differences in the uropygial gland, and what the primary differences were between olfactory cues.

## 2. Materials and Methods

### 2.1. Birds and Sampling

The animal treatment and welfare protocol listed below has been approved by the Sichuan Agricultural University Animal Ethical and Welfare Committee, ethic code is 20190035. The Nonghua strain ducks used in this study were provided by the waterfowl breeding farm of Sichuan Agricultural University. The study was conducted from August 2020 to January 2021. A total of 40 healthy ducks, including half male and half female ones with similar body weight, were reared together in the floor-reared system with 5 cm-thick sawdust bedding covering the concrete floor. The stocking density was 1 duck/m^2^. The temperature of the ducks’ room was maintained between 20 and 30 °C. The ducks were fed a standard growth period and layer duck period diet throughout the trial (Appendix A). At 20 weeks of age the ducks had reached primiparous and 6 male and 6 female ducks were randomly selected for exsanguination. The secretion of the uropygial glands was collected from the left side of the uropygial gland. Finally, the samples were kept in dry ice and then were sent to Suzhou Panomick Biopharmaceutical Technology Co., Ltd. for metabonomic analysis.

### 2.2. Metabolite Extraction

Samples were thawed at 4 °C, transferred 100 mg into 2 mL centrifuge tubes, then 600 µL 2-chlorophenyl alanine (4 ppm) of methanol (−20 °C) was added and shaken for 30 s. This is followed by 100 mg glass beads being added and put into the tissue grinder, ground for 90 s at 60 Hz. After ultrasound at room temperature for 10 min and being centrifuged at 4 °C for 10 min at 12,000 rpm, the supernatant was filtered through a 0.22 µm membrane to obtain the prepared samples for liquid chromatograph-mass spectrometer (LC-MS). A quantity of 20 µL from each sample was taken to generate the quality control (QC) samples, and the rest were used for the LC-MS analysis [27].

### 2.3. Chromatographic Conditions

Chromatographic separation was accomplished in a Thermo Ultimate 3000 system (UltiMate 3000, Thermo Fisher Scientific Waltham, MA, United States) equipped with an ACQUITY UPLC^®^ HSS T3 (150 × 2.1 mm, 1.8 µm, Waters) column maintained at 40 °C. The temperature of the autosampler was 8 °C. The mobile phase is 0.1% formic acid water—0.1% formic acid acetonitrile (positive ion mode) or 5 mM ammonium formate water—acetonitrile (negative ion mode). Injection of 2 μL of each sample was done after equilibration. An increasing linear gradient of solvent B (*v*/*v*) was used as follows: 0–1 min, 2% formic acid acetonitrile/acetonitrile; 1–9 min, 2–50% formic acid acetonitrile/acetonitrile; 9–12 min, 50–98% formic acid acetonitrile/acetonitrile; 12–13.5 min, 98% formic acid acetonitrile/acetonitrile; 13.5–14 min, 98–2% formic acid acetonitrile/acetonitrile; 14–20 min, 2% formic acid acetonitrile (positive ion mode) or 14–17 min, 2% acetonitrile (negative ion mode) [27].

### 2.4. Mass Spectrometry Conditions

The electrospray ionization mass spectrometer (ESI-MSN) experiments were executed on the Thermo Q Exactive Plus mass spectrometer (Q Exactive HF-X, Thermo Fisher Technologies, Shanghai, China) with the spray voltage of 3.5 kV and −2.5 kV in positive and negative modes. Sheath gas and auxiliary gas were set at 30 and 10 arbitrary units. The capillary temperature was 325 °C. The analyzer scanned over a mass range of *m*/*z* 81–1000 for a full scan at a mass resolution of 70,000. Data-dependent acquisition (DDA) MS/MS experiments were performed with a high-energy collision dissociation (HCD) scan. The normalized collision energy was 30 eV. Dynamic exclusion was implemented to remove some unnecessary information in MS/MS spectra [28].

### 2.5. Qualitative and Quantitative Analysis of Metabolites

The obtained raw data were converted into mzXML format (xcms input file format) through ProteoWizard software (v3.0.8789) [29]. The XCMS package of R (v3.3.2) was used for peak identification, peak filtration, and peak alignment. The main parameters were bw = 5, ppm = 15, peakwidth = c (5, 30), mzwid = 0.015, mzdiff = 0.01, method = “centWave”. A data matrix was obtained, including mass to charge ratio (*m*/*z*), retention time, peak area (intensity), and other information; these precursor molecules were obtained according to positive ion mode and negative ion mode, then these data were exported to Excel for subsequent analysis. Finally, the data of different magnitudes were analyzed and batch normalization of peak area was performed on the data. The qualitative metabolite analysis firstly confirmed the precise molecular weight of the metabolite (molecular weight error < 30 ppm), and then the MS/MS product ion spectrum of the metabolites were matched with the structural data acquired from the Human Metabolome Database (HMDB), Metlin, Massbank, Lipymaps, Mzclound, and the self-built Standard Product Database (http://query.biodeep.cn/, accessed on 4 September 2021), using the Mass Fragment software (2022 Waters, Shanghai, China).

### 2.6. Differential Metabolite Screening and Functional Analysis

We usually use the VIP (variable influence on projection) value to measure the intensity of the influence of the expression of each metabolite on the discrimination of each group of samples, thereby assisting the screening of marker metabolites. The sum of the squares of all VIP values equals the total number of variables in the model, so the average value is 1. When the VIP of a variable is >1, the variable is important. Differential metabolite screening criteria are *p*-value ≤ 0.05 and VIP ≥ 1. Among the total differential metabolites, 139 differential metabolites were annotated. First, 80 extremely significant differential metabolites were selected according to (*p* < 0.01), and then 24 volatile differential metabolites were screened according to boiling point (50–260 °C). Based on the thermal map cluster analysis, to narrow the range, 5 extremely significant volatile differential metabolites were screened out according to these 24. The PHEATMAP package in R (v3.3.2) [30] was used to perform agglomerative hierarchical clustering on each data set. Based on the MeTPA database [31], the differential metabolites were enriched with the Kyoto Encyclopedia of Genes and Genomes (KEGG), and we analyzed the metabolic pathways related to the differential metabolites in each group.

## 3. Results

### 3.1. The Chemical Components of the Uropygial Glands Can Help Distinguish the Ducks with Different Genders

After chromatographic separation, the outflow components continued to enter the mass spectrometry. Mass spectrometry continuously scans these components and collects data. One mass spectrometry was obtained for each scan, and the ions with the highest intensity in each mass spectrometry were selected for continuous description. The spectrum was obtained by taking the ion intensity as the ordinate and time as the abscissa (Figure 1). Under the conditions of positive and negative ions, there are significant differences in the mass spectrometry of the uropygial gland secretions in female ducks and male ducks. Most of the chromatographic peaks of the male ducks are significantly higher than that of the female ducks. These results indicated that sex differences greatly influence the metabolites’ composition of the uropygial glands in ducks.

We performed quality control steps during the mass spectrometry-based metabolomics to obtain reliable and high-quality metabolomics data. Through the principal component analysis (PCA) plot (Figure 2A), all QC samples were gathered, indicating the data repeatability is reasonable and the LC-MS system was stable. Then, QC data normalized the raw data to omit the batch effects.

This study performed the orthogonal projections to latent structures discriminant analysis (OPLS-DA) method on the two groups of samples. The OPLS-DA analysis can show clear differences between groups (Figure 2B). The OPLS-DA score plots showed that the male and female duck samples were significantly separated in the positive and negative ion modes. The OPLS-DA model parameters in positive ion mode are R^2^Y = 0.999, Q^2^ = 0.886; in negative ion mode they are R^2^Y = 1, Q^2^ = 0.862. The R^2^Y and Q^2^ values were close to 1.0 in the positive and negative ion modes, and the gap was less than 0.2 (Table 1), indicating that the OPLS-DA model had good accuracy and reliable predictive power.

### 3.2. Screening Differential Metabolites in Duck Uropygial Gland between Two Sexes

The LC-MS helped identify 11,311 metabolites in duck uropygial secretions in positive ion mode and 14,321 in negative ion mode. Then, the differential metabolites were screened under the standards of *p*-value ≤ 0.05 and VIP ≥ 1 (Figure 3A). Under the positive ion mode, 3831 differential metabolites, with 1493 up-regulated and 2338 down-regulated, have been screened in the duck uropygial gland between the two sexes (Figure 3B). In the negative ion mode, there were 4510 different metabolites, of which 1583 were up-regulated, and 2927 were down-regulated (Figure 3B). Because the standard products in our information database are limited, some cannot be annotated. We have only a small fraction of metabolites annotated. By annotating all metabolites, a total of 139 differential metabolites were annotated, of which 49 were up-regulated, and 90 were down-regulated (Figure 3C, Appendix A). The relative value of the metabolites of all the differential annotated metabolites was used to perform a hierarchical cluster analysis. It can be seen that the annotated metabolites were divided into 17 main sub-clusters based on their relative content (Appendix A), implying a functional difference among these metabolites.

### 3.3. KEGG Enrichment Based on the Differential Annotated Metabolites

The KEGG database used 139 differential metabolites in the duck uropygial gland between the two sexes for enrichment pathways. The results showed that a total of 77 pathways had been enriched (Appendix A). Here, we listed the top 25 significantly enriched metabolic pathways (Figure 4), including amino acid metabolism, lipid metabolism, carbohydrate metabolism, signaling molecules and interactions, and nucleotide metabolism, etc. Each of the following pathways has at least four differential metabolites. They included the ABC transporter, purine metabolism, pyrimidine metabolism, arginine, and proline metabolism; the neuroactive ligand-receptor interaction, β-alanine metabolism, aminoacyl-tRNA biosynthesis, fatty acid biosynthesis, saturated fatty acid biosynthesis, alanine, aspartic acid, and glutamate metabolism; and the glutathione metabolism, cysteine, and methionine metabolism; and the tyrosine metabolism. The following pathways enriched at least two differential metabolites, including the calcium signaling pathway, taurine, and hypotaurine metabolism; the sphingolipid metabolism, pantothenic acid, and coenzyme A biosynthesis; pyruvate metabolism, phenylalanine, tyrosine Acid, and tryptophan biosynthesis; galactose metabolism, ascorbic acid, and aldonic acid metabolism, niacin and niacinamide metabolism; neomycin, kanamycin, and gentamicin biosynthesis; and the tryptophan metabolism.

### 3.4. The Enrichment of Volatile Substances Potentially as Olfactory Cues in Duck Uropygial Gland for Chemical Signals

The volatile substances are easily emitted into the environment and can be considered candidates for the potential olfactory cues that respond to chemical signals. Among the 80 differential metabolites that reached an extremely significant difference (*p* < 0.01), we enriched the volatile substances according to the boiling point of each differential metabolite (50–260 °C) between the two sexes. We identified 24 volatile substances (Table 2) and performed a heat map analysis on them (Figure 5). Using heatmaps, we can visually judge the differences between samples/groups by the shades and differences of colors. Combined with the significance results of the statistical tests, the direction of significance can be assessed. Furthermore, we further demonstrated that five volatile substances could distinguish the sex groups of ducks very well. They included picolinic acid, 3-hydroxypicolinic acid, indoleacetaldehyde, 3-hydroxymethylglutaric acid, and 3-methyl-2-oxovaleric acid (Figure 6). All these substances are significantly higher in males than in females.

## 4. Discussion

It is well-known that the lipid substances secreted by the uropygial glands are beneficial for protecting feathers. The uropygial gland may play an essential role in maintaining the integrity of feathers. It was generally believed that birds usually peck the secretions from the uropygial gland with their beaks and smear it on feathers during preening to increase waterproofing and resistance to pathogens [32]. In our study, many chemical compounds were identified in the uropygial gland secretions. According to the metabolites annotated, lipids accounted for 12.245% of the total annotated metabolites, supporting the theory that the uropygial glands can secrete large amounts of lipids related to their protective function [33].

The uropygial gland is a special skin derivative specific to birds. Leclaire et al., found that uropygial gland secretions were significantly different between females and males. They inferred that uropygial gland secretions might be important clues for mating among birds [34]. Moreover, according to Hirao’s study, the mating frequency of males with females that had uropygial glands was significantly higher than that with females with the uropygial glands removed [19], which indicates that uropygial gland secretions may affect mating and mate selection among birds. Among the 80 differential metabolites with extremely significant differences in our study, the volatile substances accounted for 31.6%. The volatile substances were more easily emitted into the environment, and a high ratio of volatile substances accounted for the total differential substances between male and female ducks, which implied a potential role for these volatile substances to act as sex pheromones. Zhang et al. also observed that the proportion of volatile octadecyl alcohol, nonadecanol, and eicosanol in the uropygial gland secretion of male parrots was four-times higher than that of females, and proved that 3-alkanols compounds were the main substances that were transmitting sexual signals between male and female parrots [35]. Although there is some difference in the compositions of volatile substances identified in our study, both have identified alcohols, supporting that these volatile substances secreted by the uropygial glands might act as sex pheromones and transmit sexual signals between the sexes.

In our results, the uropygial gland secretions of male ducks were significantly more than those of females, including picolinic acid, 3-hydroxypicolinic acid, indoleacetaldehyde, 3-hydroxymethylglutaric acid, and 3-methyl-2-oxovaleric acid. Stralendorff et al., found that 3-methyl-2-oxovaleric acid is a broth-flavored ingredient, and its concentration in male urine is about 20 times that of female urine [36]. It can also represent the typical odor of male tree shrews (*Tupaia belangeri*). Moreover, these substances can be used as pheromones for signal communication between the sexes. Some studies have shown that indole acetaldehyde is a decomposition product of tryptophan and can attract lacewings [37]. Picolinic acid is a demethylated analogue of trigonelline. Poulin studies have shown that trigonelline can remind mud crabs to notice the presence of their natural enemy—blue crabs [38]. These studies provided necessary evidence supporting the claim that this differential volatile substance might be the candidate for the sex pheromone responsible for information exchange between the sexes.

We performed a KEGG enrichment analysis on the above five volatile metabolites and found that two were enriched. The pathways were enriched, including valine, leucine, isoleucine biosynthesis, valine, leucine, and isoleucine degradation, and tryptophan metabolism. The three enriched metabolic pathways are all an amino acid metabolism, which may be the main mechanism for determining the gender difference of the uropygial gland. Additionally, studies have shown that androgens and estrogens can promote pheromone’s contents in mouse urine, such as 2-heptanone and R, R-dehydroexotropin [39]. Asnani et al. [40]. measured the regulatory effects of hormones on the uropygial glands of male adult pigeons. They found that adrenal steroids are mainly involved in the regulation of uropygial glands. There are also studies showing that testosterone can affect the secretions of the uropygial glands in birds [41]. In the chicken uropygial gland, the expression of pro-opiomelanocortin (POMC) and melanocortin receptor 5 (MC5-R) have been reported [42,43]. Although no sex hormones have been identified as significantly differential metabolites in our study, we demonstrated a significant difference in the secretion of duck uropygial glands between the two sexes, which may because of the regulatory effects of sex hormones between males and females. As a result, the pathways of these amino acid metabolites may also be disturbed by sex hormones, resulting in changes in the content of related substances between sexes.

## 5. Conclusions

In summary, the study, which combined LC-MS and non-targeted metabolomics, revealed that duck uropygial glands secreted substances (including volatile substances) that are very different between the sexes. Picolinic acid, 3-hydroxypicolinic acid, indoleacetaldehyde, 3-hydroxymethylglutaric acid, and 3-methyl-2-oxovaleric acid in the uropygial gland secretions of male ducks were significantly higher than in those of females, which suggests that they are the candidates for a pheromone to information exchange between the sexes. These substances are essential clues for mate selection and mating among birds. In addition, the KEGG analysis showed three amino acid metabolism pathways, including valine, leucine, and isoleucine biosynthesis; valine, leucine, and isoleucine degradation; and the Tryptophan metabolism, led to changes in related metabolite levels. These metabolite pathways may be disturbed by sex hormones and are the main mechanism that determines the sex difference in the uropygial glands. Therefore, our research results laid a foundation for future research on whether uropygial gland secretions affect ducks’ reproduction and production levels.

## Figures and Tables

**Figure 1 animals-12-00413-f001:**
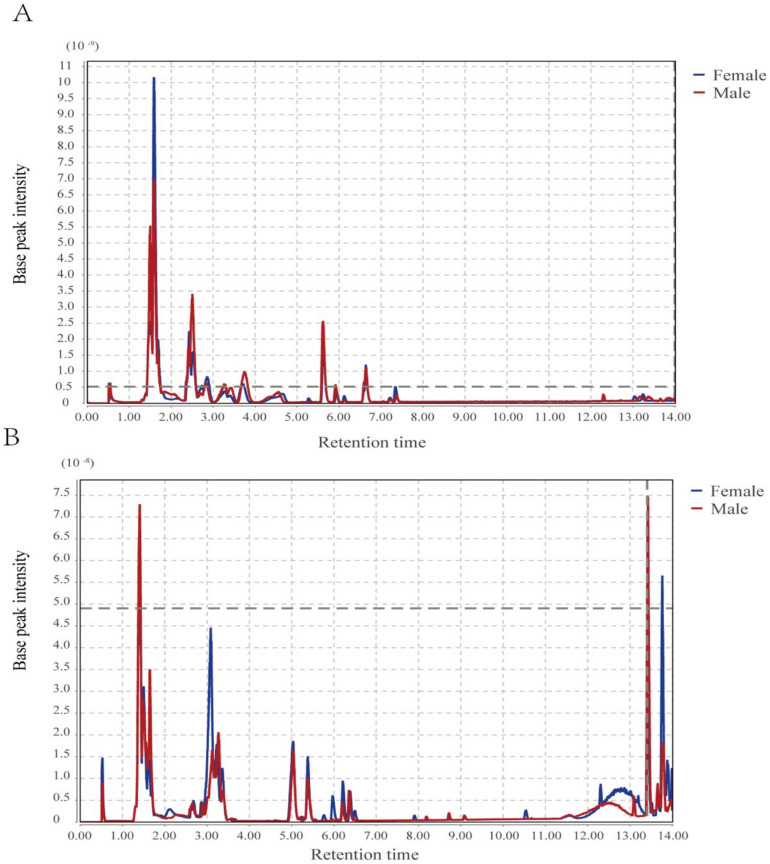
Base peak chromatogram: (**A**) base peak chromatogram in positive ion mode, (**B**) base peak chromatogram in negative ion mode.

**Figure 2 animals-12-00413-f002:**
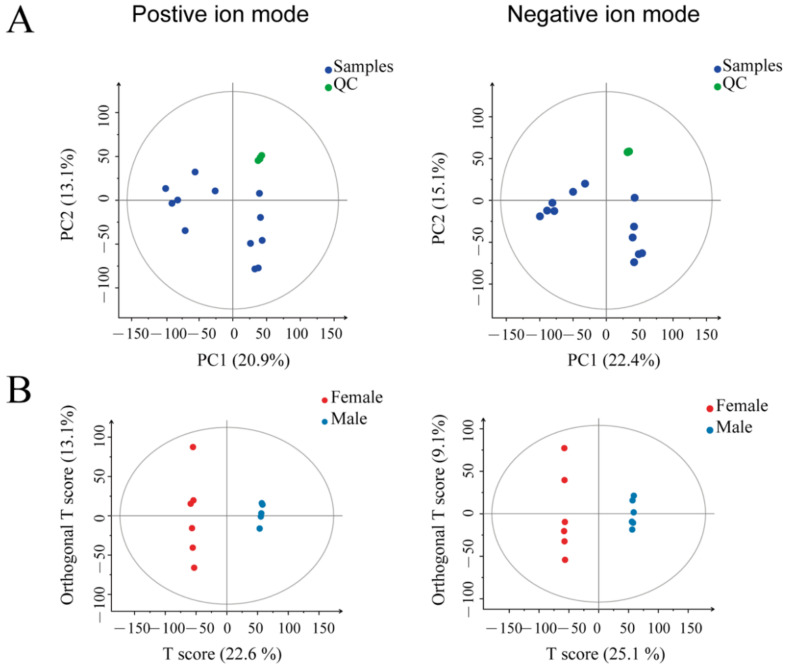
Clusters of the samples based on the relative contents of all metabolites: (**A**) PCA of samples under the positive and negative ion mode, respectively. The green dots represent the quality control (QC) samples, and the blue ones represent the experimental samples; (**B**) OPLS-DA is based on the normalized metabolomics data under the positive and negative ion modes.

**Figure 3 animals-12-00413-f003:**
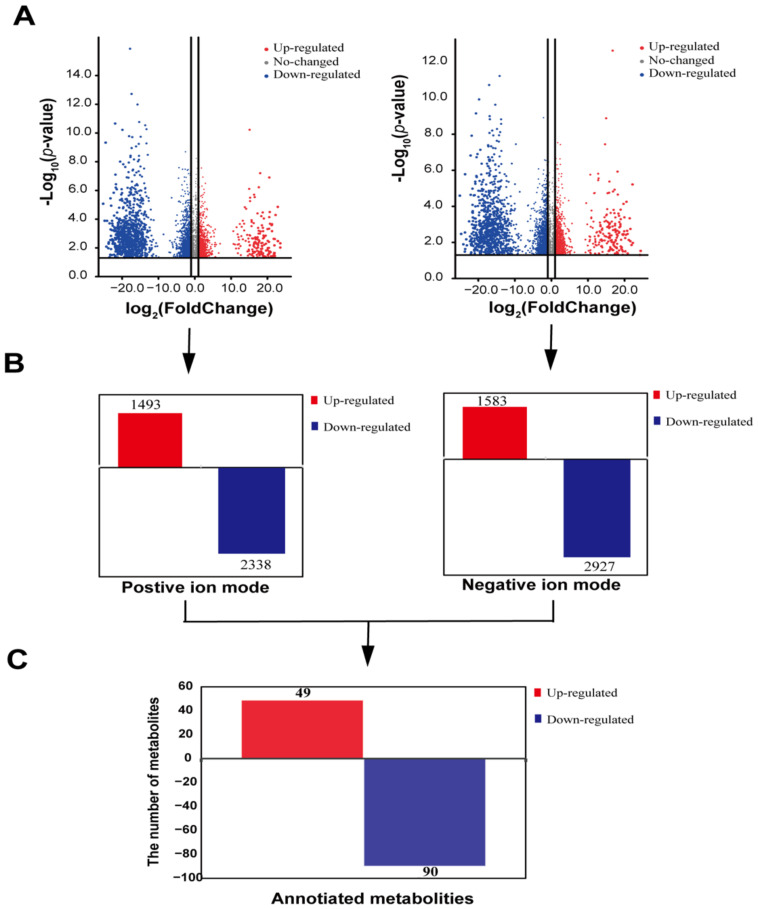
The enrichment and function annotation of differential metabolites between the sexes: (**A**) the volcanic map was used to show the whole of the differential metabolites in positive and negative ion modes; (**B**) annotated histogram of differential metabolite statistics. The red block represents the up-regulated metabolites, and the blue block represents the down-regulated metabolites; (**C**) the histogram was to show the differential annotated metabolites between sexes.

**Figure 4 animals-12-00413-f004:**
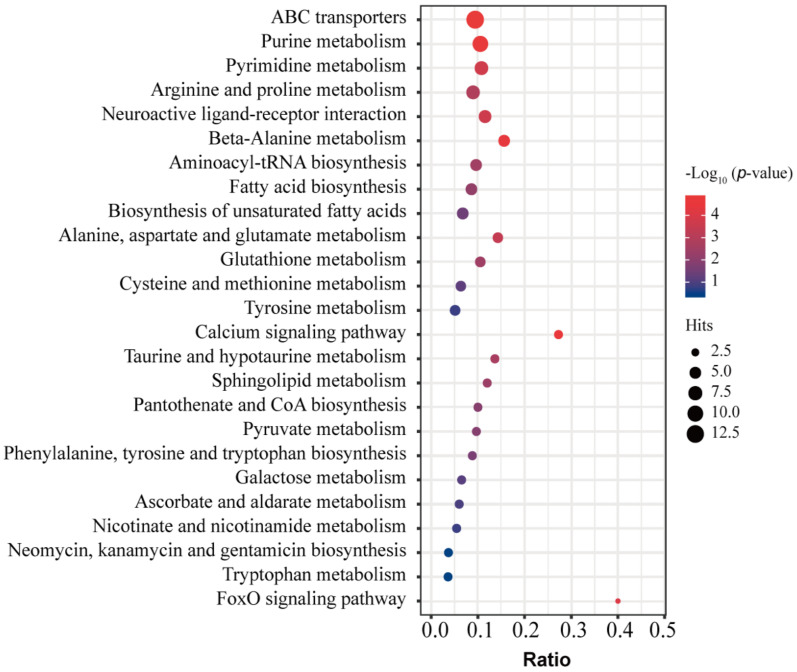
The top 25 significantly enriched KEGG pathways based on the differential metabolites in the uropygial gland between male and female ducks. The dot size represents the number of different metabolites in the corresponding pathway. The larger the dot, the more differential metabolites enriched in the pathway. The ratio is the number of differential metabolites in the corresponding signal pathway to the total number of identified metabolites. Bubbles with different colors indicate the degree of significance.

**Figure 5 animals-12-00413-f005:**
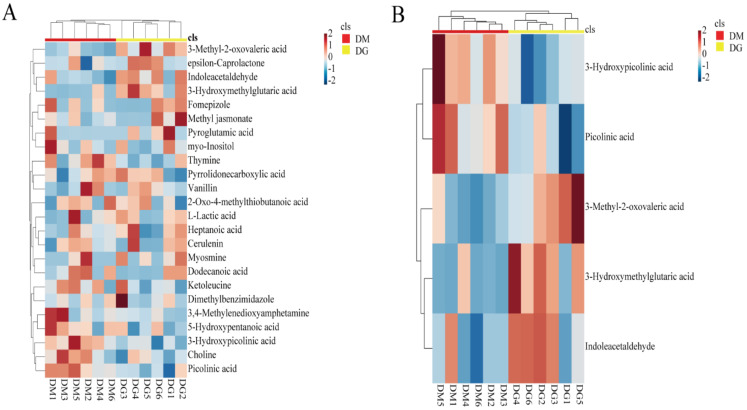
Heat map of volatile metabolites: (**A**) hierarchical clustering based on all 24 volatile substances identified in 80 significantly differential metabolites between the two sexes, (**B**) hierarchical clustering showed that five volatile substances could distinguish the sex groups of ducks. In classification, red represents DM and yellow represents DG. In legend, the color represents the relative content. The redder the color, the higher the expression level, and the bluer the color, the lower the expression level. Values are Z-score normalized for peak responses of metabolites.

**Figure 6 animals-12-00413-f006:**
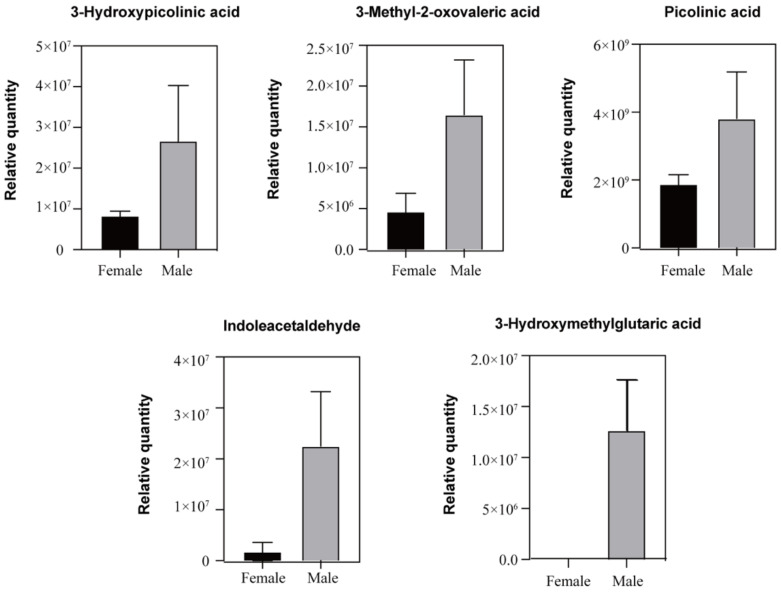
The histogram shows the differences between the sexes regarding the relative expression of the volatile secretions from the uropygial glands. All these substances are significantly higher in males than in females.

**Table 1 animals-12-00413-t001:** Evaluation parameters obtained from OPLS-DA models.

Mode	Pre	R^2^X (cum)	R^2^Y (cum)	Q^2^ (cum)
ESI+	2	0.38	0.999	0.886
ESI−	2	0.342	1	0.862

Note: ESI, electrospray ionization; OPLS-DA, partial least squares discriminant analysis; R^2^X is the interpretability of model X variable (independent variable), R^2^Y is the interpretability of model y variable (dependent variable), and Q^2^ is the predictability of the model.

**Table 2 animals-12-00413-t002:** The differential expressed 24 volatile metabolites.

Compound’s Name	VIP	Fold Change (Female/Male)	*p*-Value	FDR
Fomepizole	1.32246	2.9797 × 10^−7^	0.00278	0.02494
L-Lactic acid	1.64043	1.5124 × 10^−6^	0.00278	0.02494
Choline	1.69649	5.0626 × 10^−7^	0.00278	0.02494
epsilon-Caprolactone	1.90125	4.6829 × 10^−6^	0.00278	0.02494
Pyrrolidonecarboxylic acid	1.61869	202,830	0.00278	0.02494
Pyroglutamic acid	1.01192	480,360	0.00278	0.02494
Myosmine	1.37260	1.0198 × 10^−6^	0.00278	0.02494
Vanillin	1.61543	2.6944 × 10^−7^	0.00278	0.02494
Methyl jasmonate	1.38237	88,632	0.00278	0.02494
Picolinic acid	1.41010	0.49169	0.00508	0.02494
Thymine	1.72226	2.1228	0.00508	0.02494
Ketoleucine	1.38514	0.43895	0.00507	0.02494
Heptanoic acid	1.89661	1.7569	0.00507	0.02494
3-Hydroxypicolinic acid	1.39658	0.31057	0.00507	0.02494
Dimethylbenzimidazole	1.22882	20.875	0.00507	0.02494
2-Oxo-4-methylthiobutanoic acid	1.16255	0.07529	0.00507	0.02494
3,4-Methylenedioxyamphetamine	1.83562	0.30120	0.00507	0.02494
Dodecanoic acid	1.38491	0.15975	0.00507	0.02494
Cerulenin	1.25354	0.10284	0.00507	0.02494
5-Hydroxypentanoic acid	1.55774	1.08900	0.00824	0.03531
Indoleacetaldehyde	1.59745	0.07987	0.00824	0.03531
myo-Inositol	1.51714	4236500	0.00962	0.04094
3-Hydroxymethylglutaric acid	1.78003	2.5674 × 10^−6^	0.00278	0.02911
3-Methyl-2-oxovaleric acid	1.58333	0.28013	0.00507	0.02911

## Data Availability

The data presented in this study are available in Appendix A.

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
