# Peer review of "Metabolic Profiling Reveals That the Olfactory Cues in the Duck Uropygial Gland Potentially Act as Sex Pheromones"

_animals, 2022, doi:10.3390/ani12040413_

Round 1

Reviewer 1 Report

The paper entitled “The metabolic profiling reveals the olfactory cues in the duck 2 uropygial gland potentially act as sex pheromones" submitted for publication in Animals concerns analysis of the chemical composition of duck uropygial gland secretions and estimation of differences between male and female. The results reveal that the 25 of 80 metabolites, that reached an extremely significant difference, are volatile substances. Among them 6 substances are significantly higher in males in comparison with females, and their functions seems to be associated with the reproduction processes.

Results obtained provide an important insight into the identification of potential substances that act as sex pheromones and may be crucial olfactory clues for mate selection between birds.

The paper represents significant scientific work, and it can be accepted for publication in Animals after minor revision.

General remarks

  1. “Simple summary” needs to be written in a new way. Now, it is not acceptable.
  2. Some fragments of the paper need English correction, for instance in Materials and Methods sections 2.2 and 2.3.
  3. The authors should include some important citations in References:
    1. Amo, L., Avielé, J.M., Jarejo, D., Aránzazu, P., Rodríguez, J., Gustavo, T., 2012. Sex recognition by odour and variation in the uropygial gland secretion in starlings. J. Anim. Ecol. 81, 605e613.

  1. Karlsson, A.C., Jensen, P., Elgland, M., Laur, K., Fyrner, T., Konradsson, P., Laska, M., 2010. Red junglefowl have individual odors. J. Exp. Biol. 213, 1619e1624.

  1. Kolattukudy, P.E., Bohnet, S., Rogers, L., 1987. Diesters of 3-hydroxy fatty acids produced by the uropygial glands of female mallards uniquely during the mating season. J. Lipid Res. 28, 582e588.

  1. Whelan, R.J., Levin, T.C., Owen, J.C., Garvin, M.C., 2010. Short-chain carboxylic acids from gray catbird (Dumetella carolinensis) uropygial secretions vary with testosterone levels and photoperiod. Comp. Biochem. Physiol. B 156, 183e188.

  1. All figures should be referenced in the text (see Fig. 5 and Fig 6).

Specific remarks

Simple summary

Line 14: “As we all know” should be deleted.

Line 15: “Many animals have a special organ” should be corrected as follows: “Many species of vertebrates….”

Line 17: “unique” may be exchange with “special” or “rare”

Abstract

Line 28: “(3) Results: The results…” this phrase should be corrected

Line 33: “139 differential metabolisms” should be corrected as follows: “139 differential metabolites”

Line 39: “substances are potential substances” should be corrected

Introduction

Lines 78-87: the above citations (see point 3 of the general remarks) should be included.

Lines 98-102: please provide clear scientific hypothesis.

Results

Lines 263-271: please describe results seen in the Figures 5 and 6 (or connect described data with these Figures)

Discussion

Lines 290-292: “We speculated”: it is well-known that the lipid substances secreted by the uropygial glands are beneficial for protection of feathers.

Lines 332-336: the authors should include more information concerning involvement of hormones in the uropygial gland function:

For instance: In the chicken uropygial gland the expression of pro-opiomelanocortin (POMC) and melanocortin receptor 5 (MC5-R) have been reported (Takeuchi, S., Teshigawara, K., Takahashi, S., 1999. Molecular cloning and characterization of the chicken pro-opiomelanocortin (POMC) gene. Biochim. Biophys. Acta 1450, 452-459; Takeuchi, S., Takahashi, S., 1998. Melanocortin receptor genes in the chicken-tissue distributions. Gen. Comp. Endocrinol. 112, 220-231).

Reviewer 2 Report

This manuscript details differences in the metabolic composition of preen oil produced in the uropygial gland of male and female ducks. The results from this study provide a foundation for understanding the role of the uropygial gland in generating olfactory cues for reproductive processes. The manuscript could benefit from addressing the major comments below. Additionally, minor comments are also provided below to help improve the clarity of the manuscript.

Major Comments:

Introduction: The introduction is lacking in avian specific information regarding uropygial gland secretion, how the secretions vary by reproductive cycle and sex, and the role of olfactory secretion recognition in reproductive behavior. See the review “Female-Based Patterns and Social Function in Avian Chemical Communication” for avian specific information.

Methods: Several sections of the methods consist of imperative sentences. Additionally, the writing in the methods sections switches between past and present tense as well as first and third person.

Discussion: The discussion only focuses on the 6 volatile compounds. The other annotated compounds and the associated KEGG analysis of the annotated compounds should be discussed. Additionally, while the connection between the compounds and pheromone potential is addressed, the connection between the compounds and reproduction is unclear and needs to be strengthened.

Minor Comments:

- Introduction

Ln 46: Remove “The”.

Ln 77: “Ketones” shouldn’t be capitalized.

Ln 91: Add reference for waterfowl having more developed uropygial glands.

- Materials and Methods

Ln 114: Was the ovulatory timing considered for the females?

General: Missing information on how the volatile compounds were determined and how the 6 volatile compounds were selected.

- Results

Ln 217-218, Ln 220-224: Was there metabolite overlap between the positive and negative?

Ln 223-225: Only a small percentage of metabolites were successfully annotated. This point is not discussed.

Ln 268-269: How were these selected?

General: Fenfluramine is a therapeutic drug.

- Discussion

Ln 326: valine and tryptophan shouldn’t be capitalized.

- Tables/Figures

Table 1: There are two columns labeled R2X. From the context provided in the manuscript the 4th column should be R2Y instead.

Figure 3: Provide y-axis label for the graph in Panel C.

Figure 4: Figure legend states “Bubbles with different colors indicate different metabolic pathways”. Color indicates degree of significance.

Figure 5: Label color legend to indicate what type of values are being displayed.

Figure 6: Provide y-axis label for all graphs.

Table 2, Figure 4, Figure 5, and Figure 6 are not specifically mentioned in the text of the manuscript. Results from these figures are discussed, however.

Reviewer 3 Report

Dear authors, presented manuscript deals with an interesting and little-known topic. Below are given my comments and questions. I believe they would help you to improve your manuscript.

Abstract

The uropygial gland is a particular secretion gland of poultry. – not only poultry, most undomesticated bird species have uropygial gland

Introduction

As birds are they rely mainly on eyesight, I would mention this aspect. As a consequence, ducks have various courtship behaviors when they present their plumage, what is well described.

For insects, sex pheromones are sensitive and specific… - style of the sentence. Does you mean that insects are sensitive to pheromones or pheromones are sensitive?

In the terrestrial salamander (Plethodon shermani)… - species names are write in italics

Studies have shown that the volatile substances secreted by the uropygial glands are related to the social and reproductive activities of avians – reference

greenwood hoopoe - plase give latin name of species

Why „Ketones” are wtitten with the capital letter in the middle of the sentence?

I suggest to develop introduction, more literature, mostly birds - olfaction and pheromones.

Materials and Methods

How old were birds when you bring them? If they were chicks, does you sex chicks before you bring them to place where experiment were conducted?

Did you keep sexes together or separately?

Discussion:

It was generally believed that birds usually peck the secretions from the uropygial gland with their beaks and smear it on feathers during preening and cuticle scales… - are you certain that we can speak about cuticle scales in birds? As I know, cuticule scales are found in hairs, and birds do not have hairs

We speculated that the lipids substances secreted by the uropygial glands are beneficial for their role in protecting feathers. - this was not subject of your research, so how are you able to speculate about that in discussion?

It can represent the typical odor of male tree shrews.- please give a Latin name for species

...can attract lacewings... - the same comment as above. Please check are there given latin names for species, orders, etc.

the pathways were enriched, including valine, leucine, isoleucine biosynthesis, Valine, leucine, and isoleucine degradation, Tryptophan metabolism... - why some amino acids are written capital and others lowercase letters?

I do not see this section as proper discussion. In my opinion, this section should be expanded with more literature (birds). More examples. Are there any research about how birds sense pheromones? How your results relate to present knowledge? Can obtained results be practically applied? What is the meaning of the described differences?

Round 2

Reviewer 2 Report

No additional comments. The revisions to the manuscript are appreciated. 

Author Response

Dear Editor,

Thank you for your letter and for the reviewers' comments concerning our manuscript entitled "The metabolic profiling reveals the olfactory cues in the duck uropygial gland potentially act as sex pheromones" (Manuscript ID: animals-1516369). Thank you very much for your comments on the revision of our article, because of your comments, the quality of our articles has been greatly improved. 

In addition, the attachment uploaded below is wrong, I can't delete. I'm sorry.

Liu Hehe

Reviewer 3 Report

Please correct Latin name for Chrysoperla sinica Tjeder - Tjeder  is subspecies? Or by whom species was first described? If the second version is correct, when you give name, you should also give year, and also, the name is not written in italics (according to binomial nomenclature). However, name is not necessary to give.

Researchers have detected gender differences in the chemical composition of the uropygial gland waxes in domestic ducks before the nesting period[15] - add space bar before [15]. the same for [18], [23], [41], [43,43].

Beside this, I don’t have any comments.
